# Global discovery of human-infective RNA viruses: A modelling analysis

**Feifei Zhang** [1]*, **Margo Chase-Topping** [2,3], **Chuan-Guo Guo** [4], **Bram A. D. van Bunnik** [1,2], **Liam Brierley** [5], **Mark E. J. Woolhouse** [1,2]

1 Usher Institute, University of Edinburgh, Edinburgh, United Kingdom, 2 Centre for Immunity, Infection and Evolution, School of Biological Sciences, University of Edinburgh, United Kingdom, 3 Roslin Institute and Royal (Dick) School of Veterinary Studies, University of Edinburgh, Edinburgh, United Kingdom, 4 Department of Medicine, Li Ka Shing Faculty of Medicine, University of Hong Kong, Hong Kong, China, 5 Department of Biostatistics, Institute of Translational Medicine, University of Liverpool, Liverpool, United Kingdom

* Feifei.Zhang@ed.ac.uk

**Data Availability Statement:** All relevant data are within the manuscript and its Supporting Information files.

## Abstract

RNA viruses are a leading cause of human infectious diseases and the prediction of where new RNA viruses are likely to be discovered is a significant public health concern. Here, we geocoded the first peer-reviewed reports of 223 human RNA viruses. Using a boosted regression tree model, we matched these virus data with 33 explanatory factors related to natural virus distribution and research effort to predict the probability of virus discovery across the globe in 2010–2019. Stratified analyses by virus transmissibility and transmission mode were also performed. The historical discovery of human RNA viruses has been concentrated in eastern North America, Europe, central Africa, eastern Australia, and northeastern South America. The virus discovery can be predicted by a combination of socio-economic, land use, climate, and biodiversity variables. Remarkably, vector-borne viruses and strictly zoonotic viruses are more associated with climate and biodiversity whereas non-vector-borne viruses and human transmissible viruses are more associated with GDP and urbanization. The areas with the highest predicted probability for 2010–2019 include three new regions including East and Southeast Asia, India, and Central America, which likely reflect both increasing surveillance and diversity of their virome. Our findings can inform priority regions for investment in surveillance systems for new human RNA viruses.

## Author summary

There is a lack of evidence on the factors driving the discovery of RNA viruses in general globally. Here, we recorded the initial discovery sites of all 223 human RNA viruses and revealed its global distribution pattern. By using a machine learning method, we found that the virus discovery was driven by a combination of variables describing socio-economic level, land use, climate and biodiversity, with GDP and GDP growth found to be the two leading predictors. We also predicted the probability of virus discovery in 2010–2019 across the globe, and identified three new areas (East and Southeast Asia, India, and Central America) in addition to the historical high-risk areas. The further stratified

**Funding:** FFZ is funded by the Darwin Trust of Edinburgh (https://darwintrust.bio.ed.ac.uk/edinburgh). MEJW has received funding from the European Union's Horizon 2020 research and innovation programme under grant agreement No. 874735 (VEO) (http://www.veo-europe.eu/). The funders had no role in study design, data collection and analysis, decision to publish, or preparation of the manuscript.

**Competing interests:** The authors disclose no conflicts of interest.

analyses (distinguishing viruses transmissible in humans or strictly zoonotic, and vector-borne or non-vector-borne) helped pinpoint the explanatory factors for the discovery of specific categories of viruses and confirm the plausibility of the model. The results of our study further understanding of the spatial distribution of human RNA virus discovery, and map the likelihood of further discoveries across the world. By identifying where new viruses are most likely to be discovered in the near future the study helps identify priority areas for surveillance.

## Introduction

Since the first identification of a virus in humans—yellow fever virus in 1901—viruses have been recognised as a leading cause of human infectious diseases [1]. Numerous human diseases, from the common cold [2] to life-threatening haemorrhagic fevers [3], are caused by RNA viruses. RNA viruses such as dengue virus, norovirus, and HIV impose significant burdens on global health and the global economy [4–6]. Despite the striking declines in the incidence and mortality of RNA virus-related diseases in human following the introduction of vaccination, infections due to measles virus, yellow fever virus, and Japanese encephalitis virus continue to endanger human health and cause hundreds to thousands of deaths each year [7–9], particularly in countries with limited resources to launch mass vaccination campaigns.

Human RNA viruses comprise a total of 214 International Committee on Taxonomy of Viruses (ICTV)-recognised species as of July 2017, classified into 55 genera and 22 families [1]. Many of these—such as rabies virus, dengue virus, and measles virus—have circulated in humans for thousands of years [6,9,10], though some—such as HIV-1 and SARS coronavirus—have emerged much more recently. Typically, a virus is identified through investigation of the aetiology of a human disease (e.g. yellow fever virus [11], measles virus [12]), although some have been identified during active virus discovery programmes (e.g. rotavirus C [13], parechovirus B [14]). Viruses such as hepatitis delta virus [15] and Highlands J virus [16] were discovered by chance, as incidental findings as part of a disease investigation.

The discovery curve of human viruses, for both RNA viruses and DNA viruses, was described for the first time in 2008 [17]. Up to nine new human virus species have been detected each year since the 1950s, and this is projected to continue in coming decades [17]. The factors driving the discovery of human viruses remain to be elucidated, though two previous studies have identified predictors of the emergence of infectious diseases more generally [18,19]. In this paper, we take a spatiotemporal modelling approach to identify explanatory factors influencing the discovery of RNA viruses in humans. We assume virus discovery is determined by two underlying spatiotemporal patterns: the geographical distribution of viruses in nature, and the process of virus detection—a human activity. Geographical ranges, which vary from worldwide (e.g. Norwalk virus [4], HIV-1 [5]) to very localised (e.g. Hendra virus [20], Menangle virus [21]), are mostly determined by virus natural history, vector distribution (for vector-borne viruses), and non-human host distribution(s) (for zoonotic viruses) [22]. In contrast, virus detection reflects scientific resources and research effort [18]. An uneven distribution of research effort will lead to an uneven distribution of virus discoveries. Geographical ranges and discovery effort are likely to have different drivers [23]. Previous studies [18,24] have attempted to allow for variation in discovery effort, although this is hard to do as no direct and effective measures are available. Here, we take a different approach by identifying explanatory factors of the raw virus discovery data and then interpreting in the

discussion whether these effects might relate to virus geographic range or discovery effort or both.

## Materials and methods

### Methods overview

In this study, we followed methods and used code derived from Allen, et al [19]. We compiled and geocoded the first reports in the peer-reviewed literature of human infection for each RNA virus in our database over a period of 118 years from 1901 to 2018. A Poisson boosted regression tree (BRT) model—a method that handles spatially dependent data well—was fitted to the human RNA virus data with a set of variables thought to be potential explanatory factors. By matching the virus discovery count and all explanatory factors in each 1˚ resolution grid cell (approximately 110 km at the equator) by decade, we ranked the contribution of each explanatory factor to the predictions. We then used the parameter estimates from the best fitting BRT model to predict the probability of virus discovery for all grid cells across the globe in 2010–2019 using the values of all explanatory factors in 2015. We also conducted stratified analyses (distinguishing viruses transmissible in humans or strictly zoonotic, and vector-borne or non-vector-borne) to find the explanatory factors for the discovery of specific categories of viruses.

### Data source of human RNA viruses and updating

Data on human RNA viruses were derived from an updated version of our previously published database (https://datashare.is.ed.ac.uk/handle/10283/2970), which contains 214 viruses, with discovery dates between from 1901 to 2017. Search terms, databases searched, and inclusion or exclusion criteria for data collection was provided in our previous paper [1]. The updated version to 2018 includes nine additional human virus species recently recognised by ICTV or newly added to the database: *Nairobi sheep disease orthonairovirus*, *Achimota virus 2*, *Menangle rubulavirus*, *Madariaga virus*, *Pegivirus H*, *Central chimpanzee simian foamy virus*, *Guenon simian foamy virus*, *Enterovirus H* and *Orthohepevirus C* (**S1 Table**). The metadata provide information on discovery date, transmissibility, transmission route, and host range [1].

We defined "discovery" as the first report of an ICTV-recognised RNA virus species from human(s) in the peer-reviewed literature, and the location of initial human exposure/infection with the virus was taken as the discovery location. When the location was not given from the original paper, the site of the research laboratory was used as the discovery location (n = 3). If neither human exposure/infection location nor research laboratory site were available, the address of the first author was used as the discovery location instead (n = 19). In our database, locations of initial human exposure/infection were used for 201 (90%) viruses (**S1 Table**) and none of these were contracted while travelling. The locations were georeferenced as precisely as possible according to the original literature, ranging from precise coordinates of points to polygon-level data (e.g., city, county, district, state, or country) (see **S1 Text** for details). For unspecified locations covering more than one grid cell (**S2 Table**), sampling was used in our bootstrap framework as described below.

### Spatial explanatory factors

A set of 33 variables potentially affecting the spatial distribution of RNA virus discovery were collated and used as explanatory factors. Full details of sources, original resolutions, along with the definitions are provided in **S3 Table**. The variables were assigned to four groups: climatic,

socio-economic, land use, and biodiversity. We expect GDP, GDP growth and university count etc. to be correlated with discovery effort as they imply more resources that could be invested in virus research [25,26]. Other groups of variables including land use, climate, and biodiversity are more likely to be related to the natural geographic range of the virus [27], i.e. these variables will affect discovery via the intermediate step of emergence.

All explanatory factors and virus locations were matched by a 1˚ spatial grid cell, having rescaled or transformed the data where necessary (details of data transformation are provided in **S2 Text**). Our model matched the RNA virus discovery count in each grid cell with historical decadal climatic variables, population, GDP, and land use data (described below), so we extrapolated the data for these variables back to 1901 (see **S2 Text** for details).

## BRT modelling approach

By fitting a Poisson BRT model, we estimated the relative risk of RNA virus discovery for each 1˚ resolution of grid cell across the world as a function of the 33 explanatory factors. BRT is a tree-based machine learning method beginning to be widely used in ecological studies [28, 29]. It applies the technique of boosting to combine many simpler tree models adaptively, and renders improved predictive performance [30,31]. Tree-based learning methods are useful tools for modelling non-linear relationships and higher order interactions between variables. In addition, BRT handles spatially dependent data well, as it can capture complex structures within the data that many other modelling methods cannot [32]. We calculated Moran's I (an index of spatial dependence) to estimate the ability of the BRT model to account for spatial dependence in the virus data, using package spdep in R v. 3.5.1 (fixed distance weights were generated based on spherical distance, with the cut-off values ranging from one time to thirty times of distance of 1˚ resolution grid cell at the equator, i.e. 110km to 3300km) [33]. Unlike the traditional, significance-based approaches, BRT assesses the individual effect of each variable by estimating the relative importance of each variable to the predictions.

The bootstrap resampling approach was applied to account for spatial uncertainty in the location of virus discoveries and generated 95% quantiles. For viruses with imprecise discovery locations, one grid cell was randomly selected each time. For each grid cell with virus discovery, two grid cells with no discovery were randomly selected from all cells throughout the world that were "virus discovery free" at all time points. So, in each model, 223 grid cells with virus discovery and 446 with no virus discovery were included. We matched the virus data with all explanatory factors (using the same decade for time-varying explanatory factors, e.g. 2010 values of variables were matched with viruses discovered in 2005–2014). We assumed that the virus count in any given grid cell in each decade follows a Poisson distribution, and used the virus discovery count in each grid cell by decade as the response variable.

Using bootstrap resampling, we fitted 1000 replicate BRT models and generated relative contribution plots and partial dependence plots with 95% quantiles. The relative contribution, or the influence/weight, of each variable is an indicator of that variable's importance for predicting virus discovery counts. The relative contributions of all variables of a BRT model sum to 100%, with higher numbers indicating stronger influence on the response. We defined the most influential explanatory factors as those whose relative contributions were greater than the mean level (i.e. 100/(total explanatory factors counts*100); this study: 100/(33*100) = 3.03%) [28]. Partial dependence plots are a method of visualizing the relationships between a BRT's predictive variables and its outcome after accounting for the average effects of all other variables. The means of the predictions of all 1000 models were used to predict the probability of virus discovery across the globe in 2010–2019, using 2015 values of the 33 explanatory factors. Using the equation of Poisson probability distribution, we converted the continuous

prediction map to a probability map. We used the packages dismo and gbm in R v. 3.5.1 to fit BRT models. Parameters including tree complexity (reflecting the number of nodes in a tree), learning rate (shrinking the contribution of each added tree), and bag fraction (specifying the proportion of data to be selected at each step) were set following Elith et al. [31] to make sure each resampling model contained at least 1000 trees. The final parameters of the optimal model had the following values: tree complexity = 5, learning rate = 0.003, bag fraction = 0.5. A cross-validation stagewise function was used to identify the optimal number of trees in each model [31]. With these parameters, the 1000 replicate BRT models fitted a mean of 1214 trees.

The model's predictive performance was assessed by calculating the deviance of the boot-strap model, as well as by conducting 50 rounds of ten-fold cross-validation. Details of model validation are provided in **S3 Text** and **S4 Table**.

We also performed sensitivity analyses by i) using data from 1980 to 2000 only (as explanatory variables are available without extrapolation only for this period), and ii) removing the 22 discovery reports that were not locations of infected humans (as these are less precise). Model parameters are provided in **S5 Table**.

## Stratified analysis

Two stratified analyses were conducted to find explanatory factors specific to discoveries of different categories of virus. The first stratified analysis distinguished 131 viruses that are strictly zoonotic (all human infections are acquired from an infection in a non-human reservoir) and the 92 viruses that can spread within human populations (i.e. are transmissible, directly or indirectly, between humans) (**S1 Table**), based on previously published data [34]. A second stratified analysis was performed separately for 93 vector-borne viruses and 130 non-vector-borne viruses (**S1 Table**). We used the same BRT modelling approach for stratified analyses as we described before, and relative contribution plots and partial dependence plots with 95% quantiles were drawn for each category of virus. Model parameters are provided in **S5 Table**. Based on stratified BRT models, predictions of discovery probability for each category of viruses in 2010–2019 were also performed by using 2015 values of the 33 explanatory factors.

All statistical analyses were performed using R software, version 3.5.1 (R Foundation for Statistical Computing, Vienna, Austria), and all maps were visualised by using ArcGIS Desktop 10.5.1 (Environmental Systems Research Institute). The world shapefile used in the study was obtained from Data and Maps for ArcGIS (formerly Esri Data & Maps, https://www.arcgis.com/home/group.html?id=24838c2d95e14dd18c25e9bad55a7f82#overview) under a CC-BY license (**S4 Text**). Raw data and supporting R scripts used to generate figures for the full model are presented in **S1 R script**.

## Results

The five regions with the highest virus count were eastern North America, Europe, central Africa, eastern Australia, and north-eastern South America (**Fig 1A**). Strictly zoonotic viruses and vector-borne viruses were mostly discovered from central Africa and north-eastern South America while transmissible viruses and non-vector-borne viruses were mostly discovered from eastern North America and Europe (**S1 Fig**). The cumulative discovery count increased slowly before 1950s, and thereafter increased at a constant rate (**Fig 1B**). There is variation for the rate of discovery by geographic region (**S1 Video**). More viruses have been discovered in North America and Europe, but the numbers have decreased in recent decades. By contrast, an increased number of viruses have been discovered in Asia. Transmissible viruses and non-vector-borne viruses showed a similar temporal pattern with the curve for all human RNA

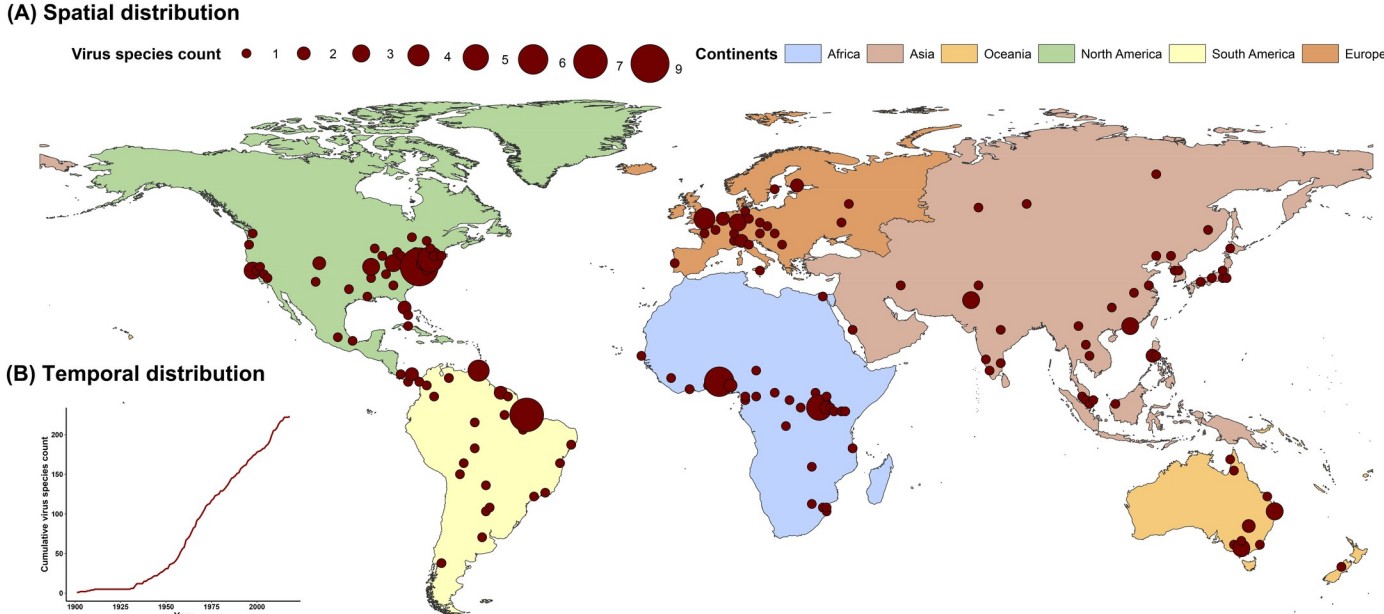

**Fig 1. Spatiotemporal distribution of human RNA virus discovery count from 1901 to 2018.** (A) Spatial distribution. The red spots indicate discovery points or centroids of polygons (administrative regions)–depending on the preciseness of the location provided by the original paper, with the size representing the cumulative virus species count. Centroid is the coordinate of the centre of mass in a spatial object. (B) Temporal distribution. The red curve indicates the cumulative virus species discovery count over time.

viruses, with an obvious increase in 1950 (**S1 Fig**). Strictly zoonotic viruses and vector-borne viruses showed a similar pattern in the early phase, with an obvious increase in 1925, but the number of new vector-borne viruses decreased after 1980 (**S1 Fig**).

Based on the full BRT model involving all 223 viruses, twelve variables had relative contributions greater than the mean (3.03%) (**Fig 2**), including two socio-economic variables (GDP growth: 12.7%, GDP: 9.9%), four variables concerning urbanization [urbanized land: 8.7%, urbanization of secondary land (i.e. the percentage of land area change from secondary land to urban land; secondary land is natural vegetation that is recovering from previous human disturbance, see **S3 Table** for details): 4.8%, growth of urbanized land area: 3.6%, and urbanization of cropland (i.e. the percentage of land area change from cropland to urban land, see **S3 Table** for details): 3.3%], five climatic variables (minimum temperature: 6.3%, precipitation change: 5.0%, latitude: 4.3%, total precipitation: 3.6%, minimum precipitation: 3.5%), and one biodiversity variable (mammal species richness: 5.1%). The partial dependence plots shown in **S2 Fig** showed the relationships between these explanatory factors and virus discovery. For the majority of explanatory factors, the relationship with discovery probability is non-linear, with large effects often seen over a narrow range of values. For example, discovery probability fell sharply if GDP growth was negative, and for very low GDP and low percentage of urbanized land; whereas it rose sharply for high minimum temperature and high mammal richness.

Our full BRT model reduced the Moran's I for the raw virus data from a range of 0.04–0.31 to 0.007–0.065 (**S3 Fig**), indicating that this modelling method with 33 explanatory factors effectively removed the spatial dependence of the model residuals. Sensitivity analyses (the analysis using data from 1980 to 2000 and the analysis after removing the 22 viruses with least certain discovery locations) revealed consistent trends with the full model, though with several changes of relative contribution.

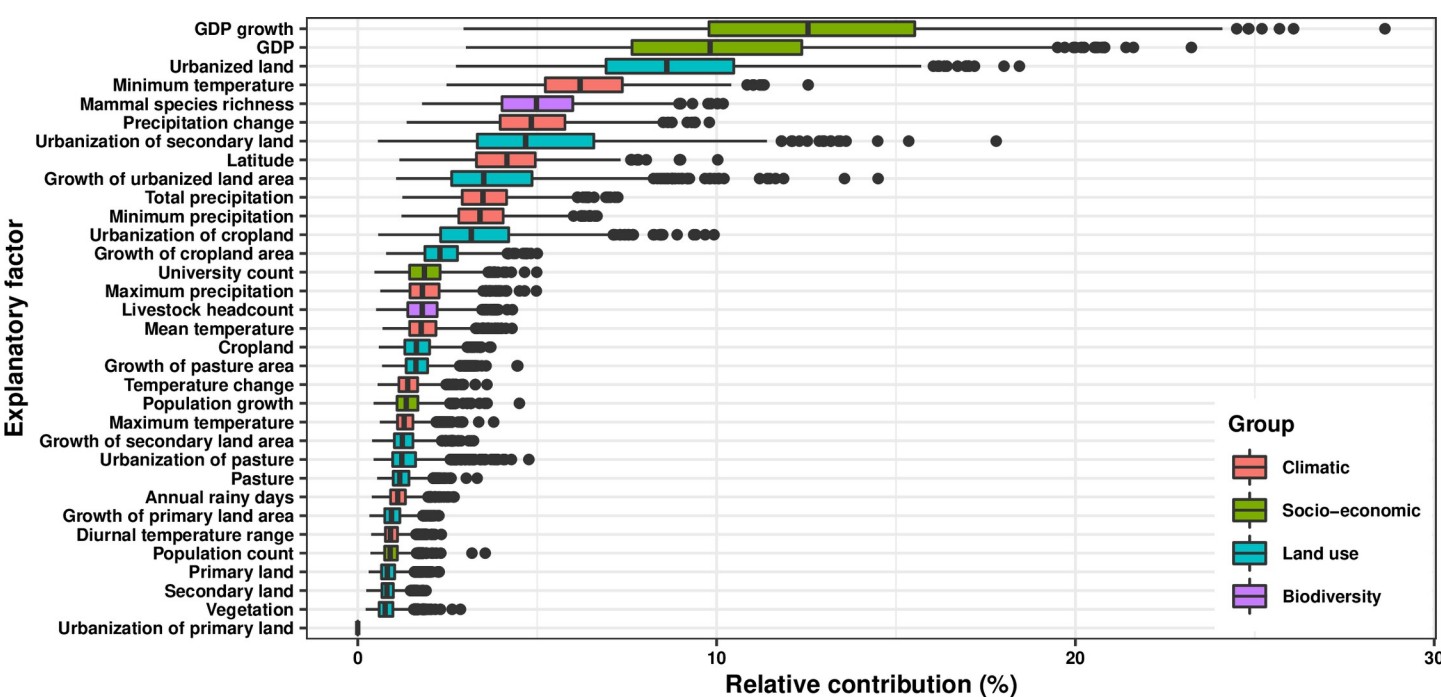

**Fig 2. Relative contribution of explanatory factors to human RNA virus discovery in the full model.** The boxplots show the median (black bar) and interquartile range (box) of the relative contribution across 1000 replicate models, with whiskers indicating minimum and maximum and black dots indicating outliers.

In the transmissibility-stratified BRT model, ten variables had relative contributions greater than 3.03% for discovering strictly zoonotic viruses (**Fig 3A**, partial dependence plots in **S4A Fig**), including four climatic variables (minimum temperature: 13.1%, latitude: 6.2%, precipitation change: 5.3%, total precipitation: 3.6%), three land use variables (urbanized land: 7.7%, urbanization of secondary land: 5.6%, growth of urbanized land area: 5.2%,), two socio-economic variables (GDP: 8.3%, GDP growth: 7.9%), and one biodiversity variable (mammal species richness: 5.6%). In contrast, eight variables had relative contributions greater than 3.03% for discovering viruses transmissible in humans (**Fig 3B**, partial dependence plots in **S4B Fig**), including four explanatory factors involving urbanization (urbanized land: 13.6%, urbanization of cropland: 9.3%, urbanization of secondary land: 6.6%, growth of urbanized land area: 3.6%), three socio-economic variables (GDP growth: 14.4%, GDP: 14.0%, population growth: 3.6%), and one climatic variable (minimum precipitation: 5.0%).

In the vector-borne-stratified BRT model, thirteen variables had relative contributions greater than 3.03% for discovering vector-borne viruses (**Fig 4A**, partial dependence plots in **S5A Fig**), including five climatic variables (minimum temperature: 17.1%, precipitation change: 7.9%, latitude: 6.2%, total precipitation: 3.8%, maximum precipitation: 3.3%), two socio-economic variables (GDP growth: 7.4%, GDP: 4.4%), one biodiversity variable (mammal species richness, 6.7%), and five land use variables (urbanization of secondary land: 4.8%, urbanized land: 4.1%, growth of cropland area: 3.7%, growth of urbanized land area: 3.6%, growth of pasture area: 3.4%). In contrast, seven variables had relative contributions greater than 3.03% for discovering non-vector-borne viruses (**Fig 4B**, partial dependence plots in **S5B Fig**), including four land use variables (urbanized land: 19.6%, urbanization of secondary land: 7.5%, urbanization of cropland: 4.5%, growth of urbanized land area: 3.5%), two socio-economic variables (GDP: 18.7%, GDP growth: 12.4%), and one climatic variable (minimum precipitation: 3.3%).

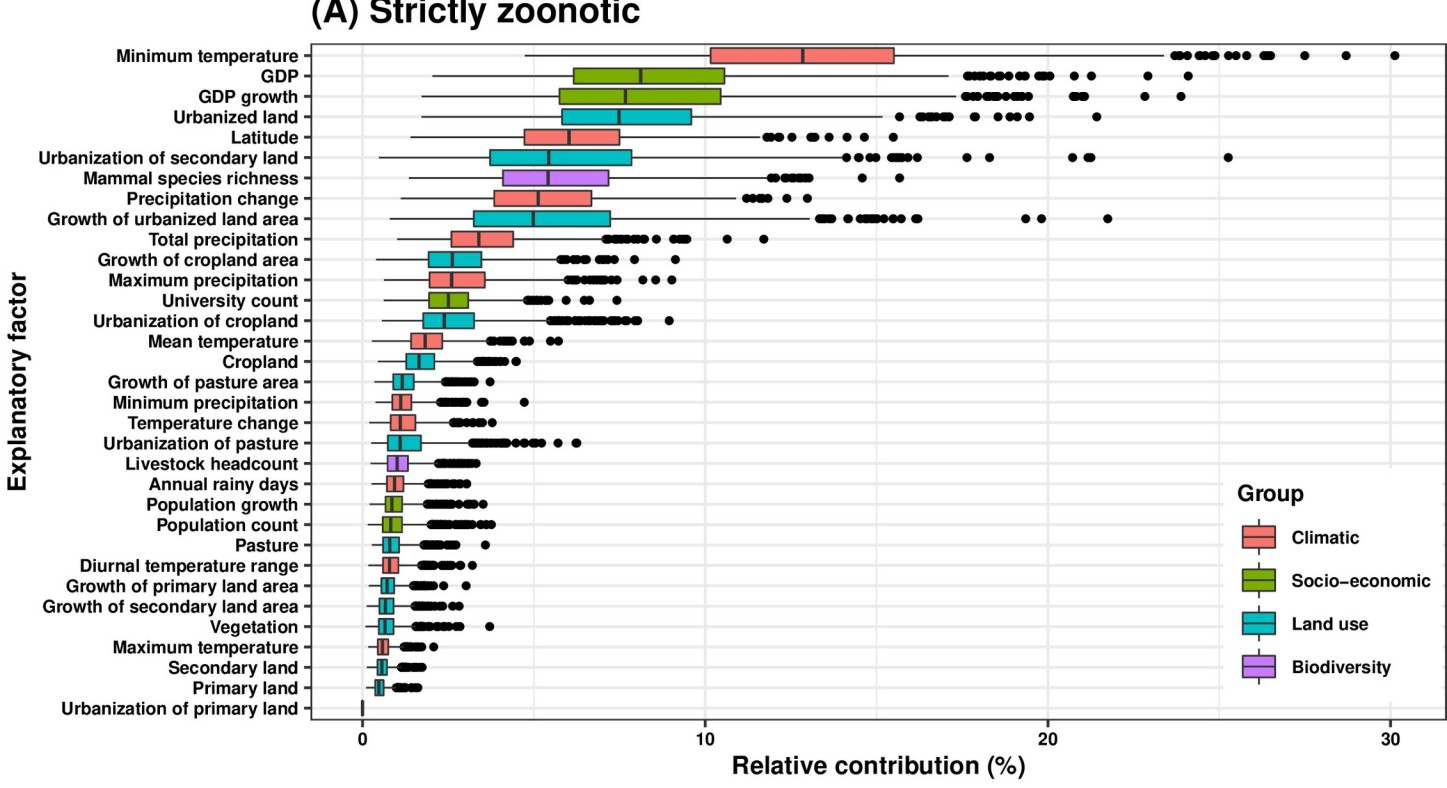

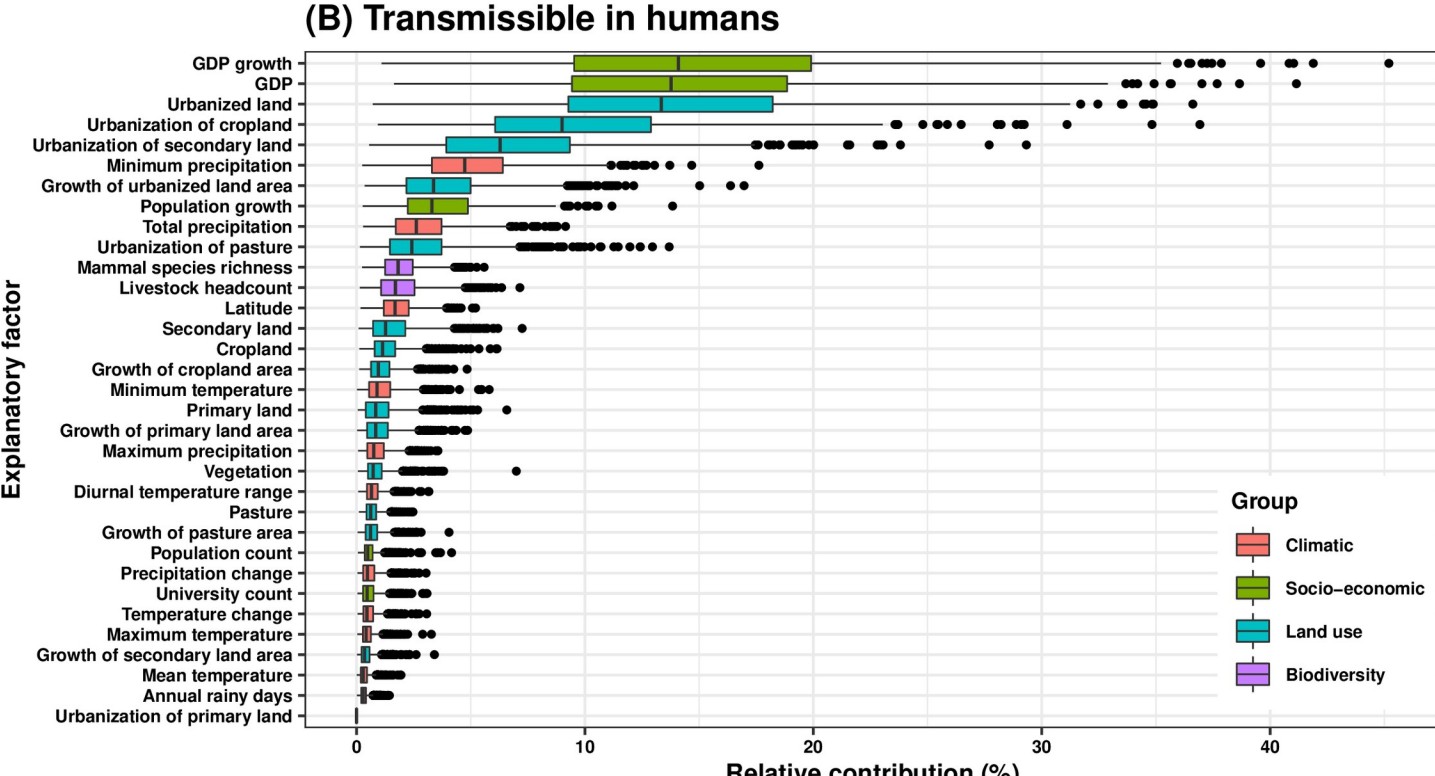

**Fig 3. Relative contribution of explanatory factors to human RNA virus discovery in the stratified model by transmissibility.** (A) Strictly zoonotic, (B) Transmissible in humans. The boxplots show the median (black bar) and interquartile range (box) of the relative contribution across 1000 replicate models, with whiskers indicating minimum and maximum and black dots indicating outliers.

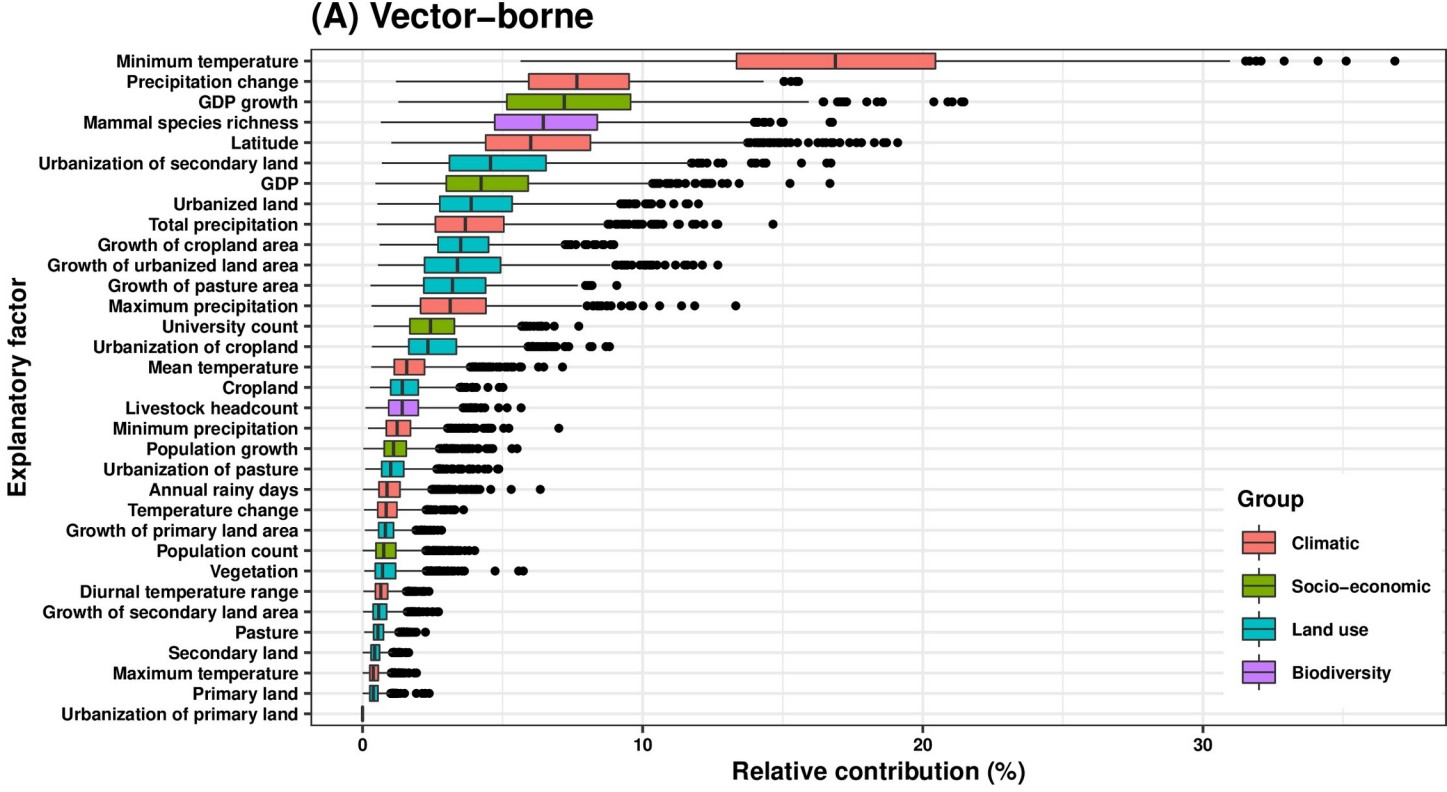

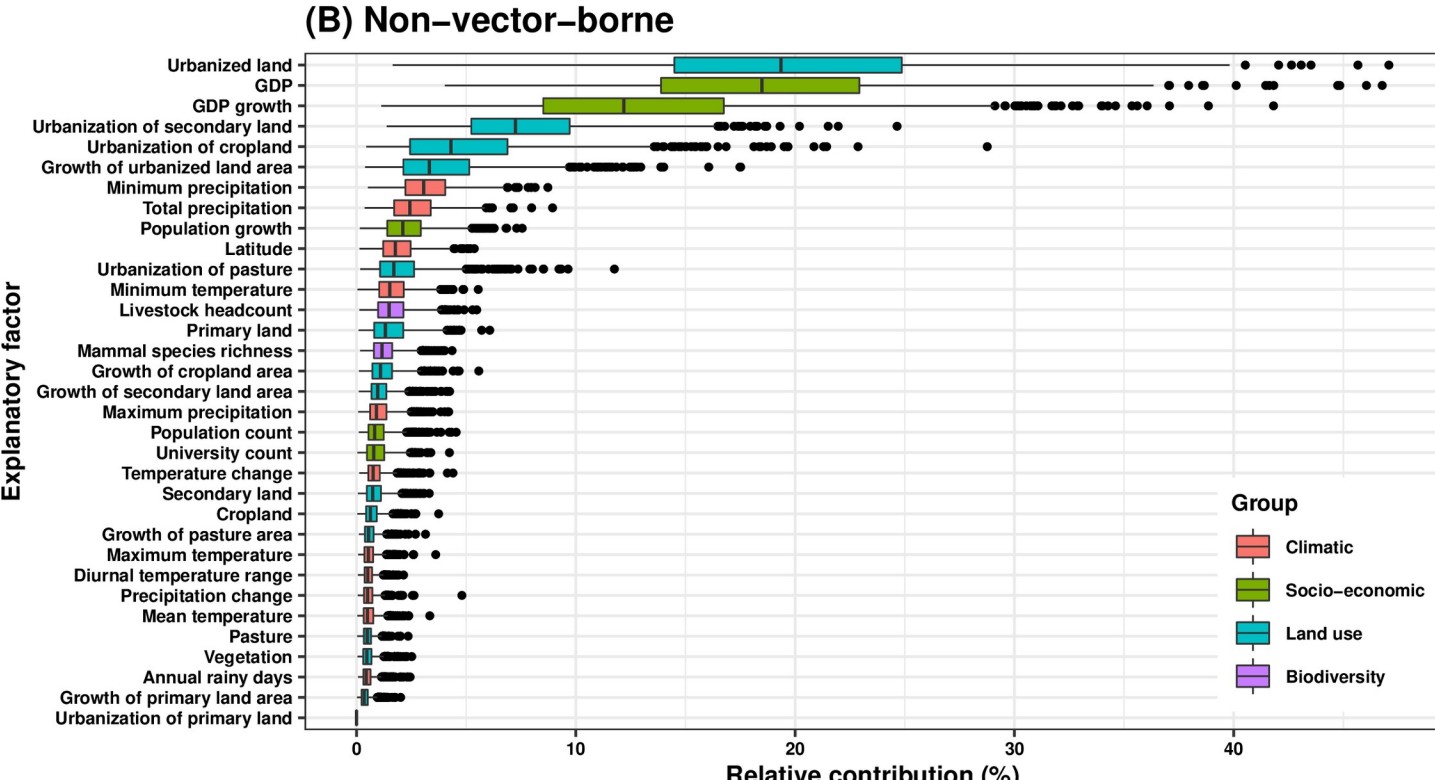

**Fig 4. Relative contribution of explanatory factors to human RNA virus discovery in the stratified model by transmission mode.** (A) Vector-borne, (B) Non-vector-borne. The boxplots show the median (black bar) and interquartile range (box) of the relative contribution across 1000 replicate models, with whiskers indicating minimum and maximum and black dots indicating outliers.

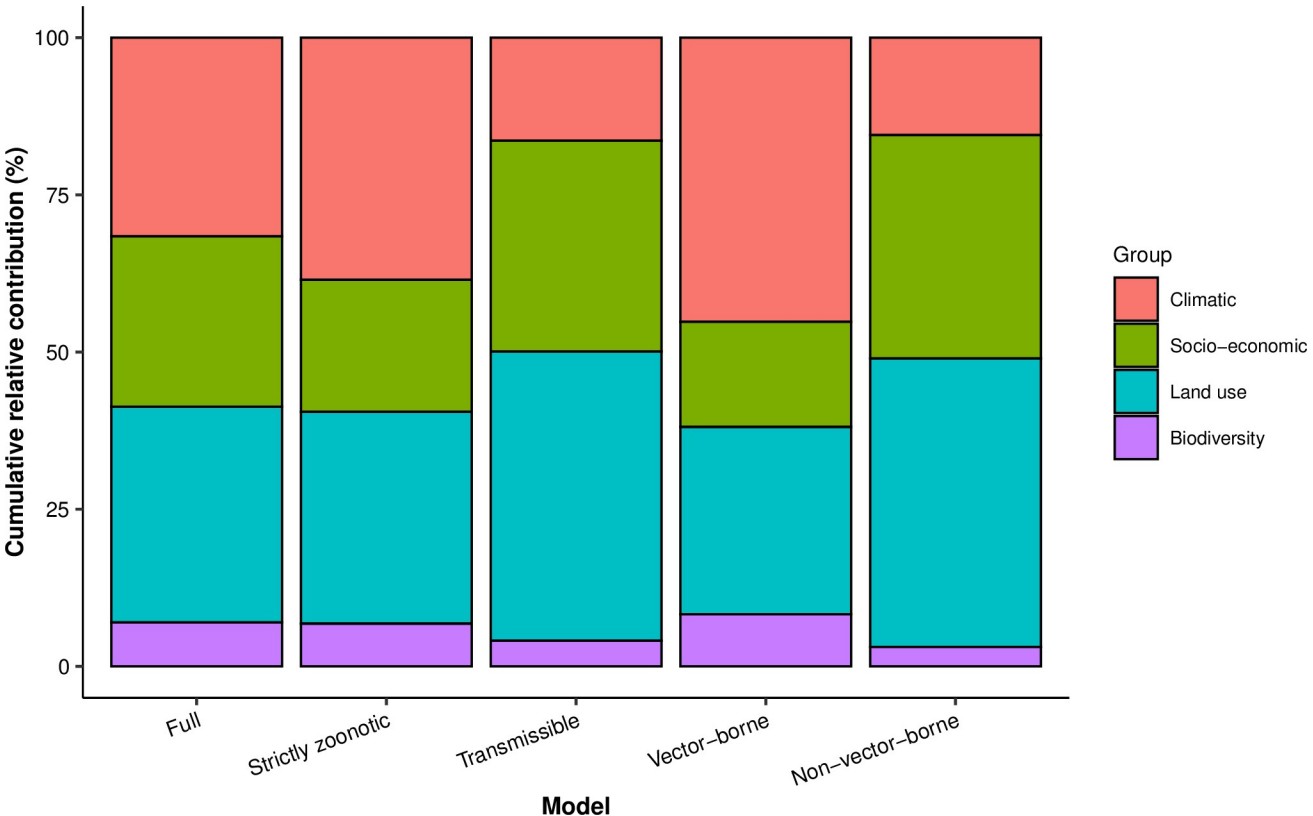

**Fig 5. Cumulative relative contribution of explanatory factors to human RNA virus discovery by group in each model.** The relative contributions of all explanatory factors sum to 100% in each model, and each colour represents the cumulative relative contribution of all explanatory factors within each group. The relative contribution of different groups to virus discovery varies across each model.

The summary of the cumulative relative contribution of each group of explanatory factors to human RNA virus discovery in each model is shown in **Fig 5**. In comparison with non-vector-borne viruses and human transmissible viruses, the discovery of vector-borne viruses and strictly zoonotic viruses is better predicted by climatic variables and biodiversity than by socio-economic variables and land use.

By applying 2015 values of all 33 explanatory factors (**S6 Fig**) to the fitted full BRT model, we obtained a predicted probability of human RNA virus discovery in 2010–2019 (**Fig 6**). Comparison with **Fig 1** indicates that virus discoveries remain relatively likely in eastern North America, Europe, central Africa, eastern Australia and north-eastern South America but, in addition, we predict high probabilities of virus discovery across East and Southeast Asia, India and Central America. All eighteen new virus species since 2010 were discovered in regions of high-risk as predicted by our model (75.0%–99.9% percentiles of predicted probability over the global range), and eleven of them were discovered in very high-risk areas (90.0–99.9% percentiles of predicted probability over the global range). The predictions of discovery for each category of virus are shown in **S7 Fig**. Broadly similar patterns as the full prediction model were seen for all four categories: high probabilities of virus discoveries are predicted in East and Southeast Asia, India, and Central America in comparison with the historical distribution (**S1 Fig**). However, there is some variation between virus categories: strictly zoonotic viruses are more likely to be discovered in northern South America, central Africa, and Southeast Asia, while transmissible viruses are more likely to be discovered in North America, East Asia, and India (**S7 Fig**); and vector-borne viruses are predicted to be more likely to be

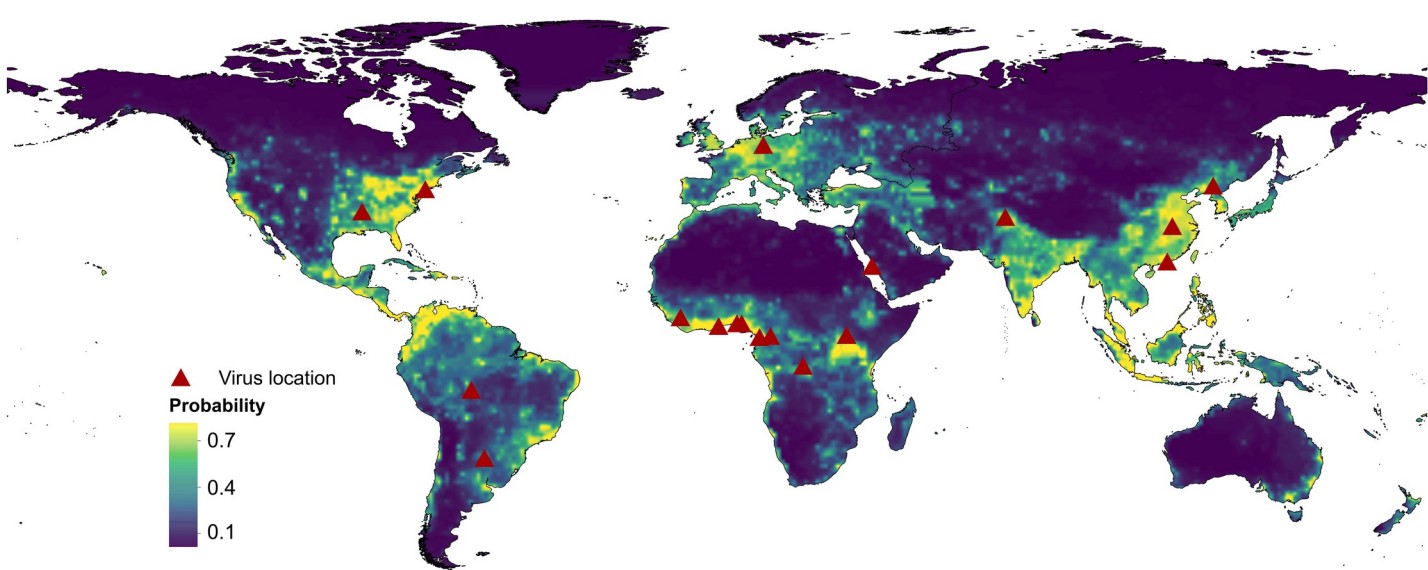

**Fig 6. Predicted probability of human RNA virus discovery in 2010–2019.** The triangles represented the actual discovery sites from 2010 to 2018, and the background colour represented the predicted discovery probability.

discovered in northern South America, central Africa, India, and Southeast Asia than non-vector-borne viruses (**S7 Fig**).

## Discussion

In this study we compiled a large body of information on global spatiotemporal patterns of human RNA virus discovery and developed a spatiotemporal modelling framework to identify explanatory factors for the discovery of new viruses. The maps of human RNA virus discovery indicate five regions with historically high discovery counts: eastern North America, Europe, central Africa, eastern Australia, and north-eastern South America. BRT modelling suggests that virus discovery is well predicted by socio-economic variables (especially GDP and GDP growth), land use variables (especially those related to urbanization), climate variables (including minimum temperature, precipitation change, latitude, minimum precipitation, total precipitation), and biodiversity (especially mammal species richness). The predicted probability map in 2010–2019 identified three new areas across East and Southeast Asia, India, and Central America in addition to the historical high-risk areas.

We focused on the discovery of RNA viruses in human(s) in this study, rather than emergence. This is determined by the attribute of the database itself, i.e. the first report of each human RNA virus from the literature review. The discovery location may or may not represent the origin of the virus. For example, HIV-1 is believed to originate from non-human primates in West-central Africa, and is estimated to have transferred to humans in 1920s [35], but the first published case from peer-reviewed literature was a Caucasian and was published by researchers in France [36].

In both the full and the stratified BRT models, GDP and GDP growth were among the top predictors of virus discovery count. This is likely to reflect that richer, more developed areas have more research funding, better access to technologies for virus detection and more effective surveillance systems. In the United States, for example, the National Institute of Allergy and Infectious Diseases (NIAID) budget on emerging infectious diseases has quadrupled over the past decades from less than $50 million in 1994 to more than $1.7 billion in 2005 [37]. Comparison of **Fig 1** with **S6 Fig** suggested that more viruses have been discovered in developed regions

with/without fast GDP growth including North America, Europe, and Australia. We note that more developed countries are more likely to first capture viruses circulating in multiple regions. Over the last 100 years, North America and Europe have witnessed a decreasing fraction of discovered viruses in more recent decades (1985–2018: 32/86 = 37%) than previously (1901–1984: 78/137 = 57%), but Asia has accounted for a higher fraction (1901–1984: 16/137 = 12%; 1985–2018: 22/86 = 26%) (S1 Video). This can be partly explained by the higher GDP and faster GDP growth in Asia in recent decades. In addition, there have also been historical hotspots in individual countries (e.g. Brazil, Nigeria and Uganda) associated with active virus discovery initiatives such as those supported by the Rockefeller Foundation [26]. More viruses are likely to be discovered in the near future in areas with high GDP growth and GDP including most of Asia (except North and Central Asia), Europe and North America.

In contrast to GDP, all other explanatory factors identified in this study appear more directly associated with virus geographic distributions, our study having the important advantage that their influence is estimated independently of GDP. We note that the relative importance of GDP is less, though still substantial, for strictly zoonotic viruses and vector-borne viruses (two large, overlapping subsets of human RNA viruses—73 out of 93 (78.5%) strictly zoonotic viruses are vector-borne) (S1 Table). This likely reflects the fact that most such viruses have geographic ranges restricted by the distributions of their vectors and/or reservoir hosts.

Consistent with this interpretation, explanatory factors related to urbanization—a consistently important category—have greatest influence for human-transmissible viruses and non-vector-borne viruses. This, again, can be explained by the fact that more viruses have been discovered in areas (especially in Asia) which have experienced rapid urbanization in recent decades (especially after 1980 [38]). Population density and growth, in contrast, are much less prominent explanatory factors, with particularly little influence on strictly zoonotic viruses and vector-borne viruses. This implies that change in habitat—from natural or rural to urban [39]—has a greater influence on virus discovery (by altering the virus geographic distributions in nature) than human population size or density.

We also found associations between the discovery of RNA viruses and climate: five of the most influential explanatory factors in the full model were minimum temperature, precipitation change, latitude, minimum precipitation, and total precipitation. That warmer and wetter climate (higher minimum temperature, more precipitation and lower latitude) is positively associated with the virus discovery is consistent with previous studies [19]. Climatic variables (especially minimum temperature) were relatively more important predictors of vector-borne and strictly zoonotic virus discovery—both these categories are more often discovered in tropical and sub-tropical regions. Forty two percent (93 out of 223) of human RNA species are vector-borne [1] and the distribution and abundance of these viruses is strongly influenced by the impact of climate on vector populations [18,40]. That climate is also relatively important for the discovery of strictly zoonotic viruses may be at least partly explained by the fact that 78.5% of strictly zoonotic viruses are vector-borne, although there may also be an association between climate and the distribution of reservoir hosts.

For biodiversity, mammal species richness was shown to make an influential contribution to human RNA virus discovery, again particularly for vector-borne viruses and strictly zoonotic viruses. Most but not all previous studies have indicated that risk of spill-over for a virus from mammal hosts to humans is positively correlated with host species richness [18,19,41] which is consistent with mammals being the main source of zoonotic viruses [34] and that as the mammal species richness increases, so does the richness of the pool of viral zoonoses [42]. Where zoonotic viruses are first discovered will be influenced, inter alia, by a range of environmental, ecological and socioeconomic factors that increase the interaction between humans and mammal reservoirs [43].

Our predicted discovery map from the full model, along with two stratified models, identified three areas—East and Southeast Asia, India, and Central America—where more viruses were more likely to be detected in 2010–2019 than have been in the past. Inspection of the historical predicted probabilities of virus discovery in **S8 Fig** indicates there has always been and is still fewer discoveries than expected in these regions. This suggests that our model is missing explanatory factors (positive or negative) relevant to these regions. However, as mentioned before, for two predicted high-risk areas—East and Southeast Asia, India—account for higher fractions in more recent times. The underlying reason may be that the explanatory factors with the greatest influence on virus discovery, such as GDP and land use variables related to urbanization, have changed substantially over time in these areas (especially China).

This study had several limitations: firstly, as indicated above, our model is missing explanatory factors (positive or negative) relevant to the three newly identified high-risk regions. Second, there is often a lag between virus discovery and publication date, though we used the latter for consistency. Third, there are other potential biases concerning spatiotemporal variation in virus detection methodologies used, and diagnostic accuracy [1]. Fourth, we used ICTV species classification following other studies [44,45], though we note that viral species for each family are defined by independent groups using different criteria, which may lead to over- or under-representation of species entries for certain families in our study compared to their phylogenetic diversity. However, we regard ICTV taxonomy as the most authoritative for comparative analysis. Last, we did not attempt to correct for reporting bias by devising a plausible metric, though previous studies have done so [18,19]. However, we explicitly included predictors that we expect to be correlated with discovery effort, e.g. GDP and university count —these are indirect and likely partial measures of effort.

The strengths of the study include use of a comprehensive data set for human RNA virus discovery, the large set of high-resolution global variables postulated to influence RNA virus discovery, and a more robust model (BRT) combining the strengths of both regression trees and boosting that is capable of solving spatial dependence. We also performed further stratified analyses (distinguishing viruses transmissible in humans or strictly zoonotic, and vector-borne or non-vector-borne) and identified differences between explanatory factors for the discovery of these specific categories of viruses. These results further understanding of the spatial distribution of virus discovery for different types, and also demonstrate that such a method can be used to identify such differences between strictly zoonotic and human-transmissible viruses or between vector-borne or non-vector-borne viruses.

In conclusion, the discovery of human RNA viruses shows both spatial and temporal variation, and is a process associated with socio-economic variables, land use, climate, and biodiversity, although the relative importance of these variables differs across different category of RNA viruses. Our study helps distinguish the relative contributions of explanatory factors reflecting the natural virus distribution and those reflecting the effort invested in virus discovery to the spatial distribution of first reports of human viruses. New human viruses are more likely to be found in areas with more rapid socio-economic growth. But the underlying geographic distribution of viruses with the potential to infect humans may be somewhat different, reflecting climate, biodiversity and changes in land use. This implies that extra investment in virus discovery in settings that are resource-poor but have other risk factors may be warranted.

## Supporting information

**S1 Fig. Spatiotemporal distribution of human RNA virus discovery count split by category from 1901 to 2018.** The map was plotted with respect to transmissibility (top left: strictly zoonotic, top right: transmissible in humans), and transmission mode (bottom left: vector-borne

viruses, bottom right: non-vector-borne viruses). In each subplot, the red spots indicate discovery points or centroids of polygons (administrative regions)–depending on the preciseness of the location provided by the original paper, with the size representing the cumulative virus species count. Centroid is the coordinate of the centre of mass in a spatial object. The red curve at the bottom left corner indicates the cumulative virus species discovery count over time.
(PDF)

**S2 Fig. Partial dependence plots for all explanatory factors that influence human RNA virus discovery in the full model.** Partial dependence plots show the effect of an individual explanatory factor over its range on the response after factoring out other explanatory factors. Fitted lines represent the median (black) and 95% quantiles (coloured) based on 1000 replicated models. Y axes are centred around the mean without scaling. X axes show the range of sampled values of explanatory factors.
(PDF)

**S3 Fig. Moran's I across different spherical distances.** The solid line and dots represented the median Moran's I value, and the grey area represented its 95% quantiles generated from 1000 samples (A: Raw virus data) or replicate BRT models (B: Model residuals).
(PDF)

**S4 Fig. Partial dependence plots for all explanatory factors that influence human RNA virus discovery in the stratified model by transmissibility.** (A) Strictly zoonotic, (B) Transmissible in humans. Partial dependence plots show the effect of an individual explanatory factor over its range on the response after factoring out other explanatory factors. Fitted lines represent the median (black) and 95% quantiles (coloured) based on 1000 replicated models. Y axes are centred around the mean without scaling. X axes show the range of sampled values of explanatory factors.
(PDF)

**S5 Fig. Partial dependence plots for all explanatory factors that influence human RNA virus discovery in the stratified model by transmission model.** (A) Vector-borne, (B) Non-vector-borne. Partial dependence plots show the effect of an individual explanatory factor over its range on the response after factoring out other explanatory factors. Fitted lines represent the median (black) and 95% quantiles (coloured) based on 1000 replicated models. Y axes are centred around the mean without scaling. X axes show the range of sampled values of explanatory factors.
(PDF)

**S6 Fig. Distribution maps for 32 explanatory factors in 2015.** The values of these explanatory variables and latitude in each grid cell were used to predict the virus discovery in the corresponding grid cell across the globe in 2010–2019. Explanatory variables were log transformed where necessary to get better visualization, not meaning they entered the model by logged values.
(PDF)

**S7 Fig. Predicted probability of human RNA virus discovery in 2010–2019 split by category.** The triangles represented the actual discovery sites from 2010 to 2018, and the background colour represented the predicted discovery probability.
(PDF)

**S8 Fig. Historical predicted probability of human RNA virus discovery by decade (except the first period with four years).** The triangles represented the actual discovery sites in each

decade, and the background colour represented the predicted discovery probability.
(PDF)

**S1 Table. Summary of the human RNA virus database.**
(DOCX)

**S2 Table. Resolution and covered grid cells for virus discovery data.**
(DOCX)

**S3 Table. List of explanatory factors included in the model.**
(DOCX)

**S4 Table. Model validation statistics for stratified analyses.**
(DOCX)

**S5 Table. Model parameters for sensitivity analyses and stratified analyses.**
(DOCX)

**S1 Text. Georeferencing human RNA virus discovery locations.**
(DOCX)

**S2 Text. Transformation of resolution for explanatory factors and data extrapolation.**
(DOCX)

**S3 Text. Result of model validation.**
(DOCX)

**S4 Text. Source and permission for the world shapefile used in the study.**
(DOCX)

**S1 Video. The spatiotemporal pattern of human RNA virus discovery.** The red spot represents the discovery location of each virus species over time. The red curve at the bottom-left corner represents the cumulative virus species count over time.
(MP4)

**S1 R script. A zipped file with the raw data and R code that was used for generating figures for the full model.**
(ZIP)

## Acknowledgments

We thank Donald Smith (University of Edinburgh, Edinburgh, UK) for validating the database, Melina Beykou and Melissa Taylor (University of Edinburgh, Edinburgh, UK) for checking the transmissibility of RNA virus, and Thibaud Porphyre (University of Edinburgh, Edinburgh, UK) for statistical guidance.

## Author Contributions

**Conceptualization:** Feifei Zhang, Margo Chase-Topping, Mark E. J. Woolhouse.

**Data curation:** Feifei Zhang, Liam Brierley.

**Formal analysis:** Feifei Zhang.

**Funding acquisition:** Mark E. J. Woolhouse.

**Investigation:** Feifei Zhang.

**Methodology:** Feifei Zhang.

**Project administration:** Mark E. J. Woolhouse.

**Resources:** Liam Brierley, Mark E. J. Woolhouse.

**Software:** Feifei Zhang, Chuan-Guo Guo.

**Supervision:** Margo Chase-Topping, Mark E. J. Woolhouse.

**Validation:** Feifei Zhang, Chuan-Guo Guo.

**Visualization:** Feifei Zhang.

**Writing – original draft:** Feifei Zhang, Mark E. J. Woolhouse.

**Writing – review & editing:** Feifei Zhang, Margo Chase-Topping, Chuan-Guo Guo, Bram A. D. van Bunnik, Liam Brierley, Mark E. J. Woolhouse.

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
