## [Decision Letter · Decision Letter 0]

29 Jul 2020

Dear Ms Zhang,

Thank you very much for submitting your manuscript "Global discovery of human-infective RNA viruses: a modelling analysis" for consideration at PLOS Pathogens. As with all papers reviewed by the journal, your manuscript was reviewed by members of the editorial board and by several independent reviewers. In light of the reviews (below this email), we would like to invite the resubmission of a significantly-revised version that takes into account the reviewers' comments.

The manuscript has significant strengths.  Many of the specific points have been addressed by the reviewers.  Here, I would like to summarize some overarching points, as well as reinforcing some of the reviewers' comments. 

The manuscript is a methodological tour de force, consonant with this group's previous work.  The approach is very interesting, building on this group's earlier work on rates and locations of pathogen discovery (e.g., their Ref. 11).  However, as the manuscript was not submitted as a methodological paper, therefore this manuscript should explicitly discuss how the work advances current knowledge.  In addition, as Reviewer 2 notes, the distinction and overlap between results from studies of (i) pathogen discovery and (ii) pathogen emergence should be clarified.  In the absence of formal analysis of reporting/publication bias, disentangling these threads becomes particularly critical. The authors themselves at times seem unsure which of these they're referring to.  For example, on p. 3, ll. 47-50, the authors state, "The results of our study further understanding of the spatial distribution of human RNA virus discovery, and maps the likelihood of further discoveries across the world. By identifying where new viruses are most likely to occur in the near future the study helps identify priority areas for surveillance.", but  in the Introduction, the authors note, quite astutely, "We assume virus discovery is determined by two underlying spatiotemporal patterns: the geographical distribution of viruses in nature, and the process of virus detection—a human activity."   In the cover letter, the authors imply that using this complementary approach can help to correct for resource bias in identification (discovery), among other things.  In that case, it should be possible to demonstrate the valued added  by this approach.  Some specific examples could be used here.  What are the significance and interpretation of these results, and are there recommendations?  If there is a close correlation of explanatory factors between studies of discovery and studies of "emergence", why is that so?  Does that imply we're looking in the right places, or simply the obvious ones, or that the factors that enhance discovery coincide with those that promote emergence, such as urbanization, travel, and increasing scientific capacity in developing countries?  Does the analysis suggest a better strategy for searching?  With an analysis this detailed one hopes it would be possible to go beyond the descriptive and begin more critically exploring mechanisms and hypotheses.

The figures are very elegant, but to this digital non-native, some seemed duplicative, and seemed to obscure the main points.  The text itself is rather dense.  For the readership of PLoS Pathogens, I think more explanation and context are required.  Similarly, the Discussion should more explicitly discuss any unique benefits and added value of this approach.  Again, an example or a few examples of how this can be used to advance our knowledge would be invaluable.  Earlier analyses appear to have arrived at similar conclusions from a different vantage point so ways to use the analyses in the manuscript to extend this knowledge beyond the descriptive would be very useful.  The analysis does study trends over time (including a video), and that would seem to merit more discussion as well.

This is admittedly a difficult subject.  The authors' group has a developed excellent database that they have generously and most commendably shared freely with all interested researchers.  As they indicate, however, the data are still sparse and rife with questions of definition.  Nevertheless, it remains a useful resource, and I hope that this manuscript can advance understanding of the mechanism of the emergence process.  The Discussion seems somewhat limited, and I would encourage the authors to consider additional conclusions and implications of their work in their discussion.

On another note, my sincere apologies to the authors for the unusually prolonged review period.  It is an unfortunate irony of timing to submit a manuscript on this subject just as an emerging viral pandemic appears, preoccupying all our colleagues.

We cannot make any decision about publication until we have seen the revised manuscript and your response to the reviewers' comments. Your revised manuscript is also likely to be sent to reviewers for further evaluation.

Sincerely,

Steve

Stephen Morse

Guest Editor

PLOS Pathogens

David Wang

Section Editor

PLOS Pathogens

Kasturi Haldar

Editor-in-Chief

PLOS Pathogens

orcid.org/0000-0001-5065-158X

Michael Malim

Editor-in-Chief

PLOS Pathogens

orcid.org/0000-0002-7699-2064

Reviewer's Responses to Questions

**Part I - Summary**

Reviewer #1: This paper uses a number of techniques, most significantly an ensemble of boosted regression trees (BRT)-based approach, to find associations between a set of predictor variables and their outcome variable representing RNA viral discovery.

The paper appears to follow methods and workflows strongly derived from Allen et. al. (2017) (currently citation 13), particularly in the generation of its BRT models and associated figures. I would suggest that the Methods section include a statement like, “We followed methods and used code derived from Allen et al (2017)…,” where describing derived methods/code/workflows.

In addition, the paper should share the code used to generate it’ models, as per the recommendations in PLOS’s Materials and Software Sharing policies (https://journals.plos.org/plospathogens/s/materials-and-software-sharing). Sharing model and figure code removes any ambiguity that may be present in the natural language description of a study’s methods as a result of space constraints or the difficulty of describing complex processes in natural language. Code should preferably be shared with an appropriate license on GitHub and assigned a DOI using a service such as Zenodo, but could also be included with the study’s Supplementary Information, as its dataset already is.

Following a similar process but with a separate outcome dataset with different inclusion criteria and a number of different predictors, the paper comes to similar conclusions as other studies on the same topic (particularly references 12 and 13) for its main model. That the different datasets would yield finds similar results undergirds the conclusions of the earlier papers. Some of their high-influence variables are similar (e.g. mammal species richness) and others are divergent (e.g. finding an association and significant impact for climate variables). These differences between similar previous studies are worth publishing.

This study also runs different BRT models for biologically distinct subsets of RNA viruses, and the finds notable differences between the drivers of these models. These findings are notable because they describe differences in the global distribution of viral discovery for different types of viruses, and second because they demonstrate that such a method can be used to identify such these sorts of differences between, say, zoonotic-only and human-transmissible diseases.

It’s my opinion that these contributions — the influence of climate variables and the stratified analyses — are the biggest points of departure from earlier literature, and this this studies best contributions.

There are a few other points that could be improved in the study’s discussion of its findings. These are noted below.

Reviewer #2: There are many strengths to this study, including their recognition of the need to deal with the critical issue of variation in research effort.

However, throughout the manuscript I found myself challenged by what ‘virus discovery’ really means in the context of this study. I was often confused about whether the authors are truly focused on the factors that influence virus discovery, or if they are actually focused on factors that influence virus emergence. Indeed, the opening two sentences of the abstract reflect this unresolved conflict between predictors of emergence (“…the prediction of where new RNA viruses will emerge is a significant public health concern”) and virus discovery (“we matched these data….to predict the probability of virus discovery”). While this might seem like semantics, I think it is critical to identify and articulate specifically what the paper is trying to achieve. Discovery and emergence are not the same process, and thus presumably should be approached differently in study design and interpretation. There are many many examples throughout the text where the two processes are confused and, unfortunately, this limited my ability to evaluate the specific contribution of this study.

I kept going back to the example of Reston virus, which was discovered in Virginia (U.S), but geocoded to Manilla, Philippines. If the study is about understanding factors underlying virus discovery, it seems odd to ascribe the discovery of Reston virus to Manilla. Reston virus was discovered because infected animals happened to end up in the hands of an ebolavirus expert at a high-containment lab, in a completely different country. I can understand why the authors decided to code Reston virus to the Philippines (it is where the virus naturally circulates); however, I question whether these spatial covariates are relevant to its discovery in the U.S. In this case, I think the spatial data that is linked to the Philippines would tell us more about virus transmission and emergence than discovery, and I worry about spurious associations. The same issue exists for Ebola virus (species Zaire ebolavirus), which was ascribed to DRC when it was actually discovered by Prof Piot in Belgium.

Ignoring, for a moment, whether these viruses are appropriately geocoded for their particular question, there is another problem. That is, the approach taken to geocode the viruses is inconsistent. For example, while Reston virus and Ebola virus were coded to reflect their point of origin (i.e., where the virus was naturally circulating), other viruses like influenza were coded to reflect where they were indeed first ‘discovered’. Influenza was first characterized in 1933 in Mill Hill, London, and the authors have geocoded it accordingly. The approach taken in coding influenza therefore appears to be different from the approach taken for Reston and Ebola. While the spatial covariates for Reston virus and Ebola virus are capturing one type of information (location of original circulation), the covariates for other viruses like Influenza or Hepatitis C are capturing a different type of information (location where the virus was actually discovered, but not necessarily or exclusively where it was circulating).

Throughout the paper, I worry that ‘discovery’ and ‘emergence’ have been confused. In the author summary, they state “By identifying where new viruses are most likely to occur in the future, the study helps identify priority areas for surveillance.” The phrase “most likely to occur” suggests they are indeed interested in emergence. Equally, in the discussion they state “This implies that it is the change in habitat—from natural or rural to urban—has a greater influence on virus discovery than human population size or density.” Again, this sentence implies that the study is about emergence, not discovery. However, elsewhere in the text they refer specifically to need to “identify the factors driving the discovery of RNA viruses”. I think it would be important to more clearly articulate exactly what the study is focused on (I think this is virus discovery) and comment on how this links or informs on our understanding of viral emergence (if indeed it does). Currently, I am not sure how to interpret the following statement in their abstract: “…areas with the highest predicted probability for 2015–2024 include new foci in East and Southeast Asia, India, and Central America”. Does this really mean areas where new viruses are likely to be discovered? Or does it mean areas where new viruses are likely to emerge, and then subsequently discovered simply because they are emerging?

I also found the definition of ‘discovery’ confusing. It was defined as ‘the first isolation of a virus in a human patient.’ Given that many human viruses were first isolated years after they were actually discovered, it would be good to clarify whether the authors really do mean ‘isolated’. (To a virologist, this means culture of the virus in vitro or vivo). In reviewing the supplemental material, I see that the authors have used 1989 for hepatitis C. This makes me think they do not actually mean ‘isolated’, but rather ‘discovered’. Hep C was first cloned in 1989, but it was not successfully grown in culture (‘isolated’) until around 2005. Again, this is inconsistent with examples like influenza. Note, I did not review the list of viruses exhaustively, so would recommend that the authors double-check everything. In addition, the term ‘patient’ is included in their definition and is perhaps misleading. Many of the viruses they have included are probably not human pathogens, or at least have not been conclusively linked with human pathology. Reston virus is a good example of this.

Reviewer #3: This paper is a timely advance for understanding discovery of human RNA viruses that will be of interest to researchers working on emerging infectious disease. The methods are rigorous and this work uses a new approach to evaluate patterns in virus detections. This work highlights important correlates of newly recognized RNA viruses that can help predict future emergence. The insights are not especially novel but findings are important and confirm findings previously reported using new data and methods. The only major concern is that the very broad presentation of correlates in a model limits the impact and relevance of this work.

**Part II – Major Issues: Key Experiments Required for Acceptance**

Reviewer #1: There are no major new experiments required for publication.

Reviewer #2: (No Response)

Reviewer #3: The analyses are thorough and evaluate a wide range of potential correlates for virus discovery. But a major limitation of this manuscript is that the inferences are not clear beyond very broad assessment of relative contributions for factors evaluated that are presented in numerous figures in the main text and SI. Concrete and meaningful functional relationships between key predictors and the outcome are not derived or explained in the results and discussion. Exploration of specific values for predictive factors and how these influence quantity of virus discovery would bring meaning to these findings. For example, the authors should explore how urbanization influences virus discovery – what levels of urbanization were more influential and what does this mean for virus discovery based on expected future trends in urbanization? Examples to highlight virus discoveries and the factors related to their first detection that underlie broad patterns in the data would bring clarity and meaning to findings.

The many figures in the main paper and SI are largely of the same variety of relative contributions to the model and probability maps – these should be narrowed down to only those that highlight major findings with improved quality. A figure that shows specific values of the one or two key factors and the predicted influence on number of viruses discovered would enhance the presentation of results.

Can the authors further explore the rate of detection over time? How did rate of detection or the temporal distribution vary by the 4 classifications of viruses; did the rate of detection vary by geographic region? The influence of the predictors on prediction is the most novel aspect of this work yet it is not explored beyond highlighting broad regions with increased probability. Can the authors add specific inferences based on the analysis in terms of the number of viruses expected to be discovered every year, by virus groups examined? Is it possible to predict this beyond 2024?

**Part III – Minor Issues: Editorial and Data Presentation Modifications**

Reviewer #1: On line 49, the authors state that they identify “where new viruses are most likely to occur”, but they are actually identifying where they are most likely to be discovered. This might seem like splitting hairs, but since the authors did not treat the effect of reporting bias separately, it’s an important distinction to make.

The authors’ discussion of this very phenomenon on lines 74–83 is cogent, and succinct. Distinguishing between virus species range and discovery probability is one of the clearest explanations of this subject I’ve read.

However, their methodological treatment of reporting bias is not clearly described here or in the methods section. In the introduction, it is described on lines 84-86: “Here, we take a different approach by identifying explanatory factors of the raw virus discovery data and then considering whether these relate to virus geographic range or discovery effort or both.” This might be made clearer by saying something like “and then interpreting in the results whether these effects might relate to virus geographic range or discovery or both.

The Discussion section notes that the paper does not attempt to explicitly correct for reporting bias (line 361). In this mention, they cite Jones et al. (ref. 12) but not the subsequent paper (ref. 13) which was an improvement on the Jones et al. method. However, the study includes variables expected to be associated with discovery effort, e.g. GDP and university count. Earlier in the discussion, it is noted that GDP growth is among the top predictors, and its effect differs across the stratified models. That the study does not attempt to factor out reporting effort means that its authors should be careful to interpret their results only as “viral discovery” and not “viral emergence”.

The paper should be given a minor copy edit (e.g. “spatial dependencies” -> “spatially dependent” on line 93).

The contribution of the paper’s use of k-means clustering to its central research questions is unclear. The paper’s initial discussion of the k-means clustering analysis suggests that this was perhaps used as selection criteria for including points in the model (line 92, “if the spherical K function detected a clustered pattern, a Poisson boosted…”). However, the flow chart in the Supplementary Information gives the impression that the clustering analysis was not part of the BRT modeling workflow, but a separate analysis. If the k-means analysis is part of the modeling workflow, its relation to the BRT models should be made more clear. If it is a separate analysis, the authors should devote space in the Discussion section to interpreting its outcome (the figure in the supplementary information is not self-explanatory) or remove it from the paper entirely.

Line 167: For readers unfamiliar with BRTs, perhaps move the sentence about partial dependence plots to a new paragraph, or introduce it in a slightly different way, something like, “Partial dependence plots are a method of visualizing the relationships between a BRT’s predictive variables and its outcome…”.

Line 199: The paper talks about making predictions for 2015-2024, but these predictions are in fact the model’s output for 2015 variables. The authors do not go into enough detail about how the variables were matched to decade for us to assess this. If the outcome of the model uses 2015 variables because these are the most recent ones available, the model’s conclusions are still valid, but stating those predictions as “2015-2024” without justification would be misleading.

Line 231: The use of Moran’s I to demonstrate the model’s removal of spatial residuals is very clear.

Figures 3 and 4 uses the labels “Relative Contribution” and “Relative Influence” inconsistently. It should use only one. Readability would be much improved if the subplots were titled, rather than just being labeled A and B, so the reader can see which group they represent rather than referring to the caption.

The maps use a color palette which is not perceptually uniform — a color palette such as Viridis or Google’s Turbo color palette would be better for displaying this kind of continuous data on a map.

Line 365: The high-resolution variables were all scaled down to a 1º grid, so the higher resolution of the source variables was not a factor in the models’ output.

Reviewer #2: The authors considered a combination of socio-economic, land use, climate, and biodiversity variables as correlates of virus discovery. What is the rationale for how/why they are relevant to virus discovery? And given that land use, climate, and biodiversity variables seem more associated with emergence, can they provide some discussion on why these were selected, beyond saying ‘it’s a way to deal with variation in research effort’.

For their future predictions – did they account for projected changes in the factors that were correlated? Forecasted changes in GDP, land use, climate, etc.

Line 61 – “Human RNA viruses”. Can they authors define what they mean by a human RNA virus? Is it a virus that was found in a human at least once? And if so, is that sufficient to call it a human virus?

Lines 62-64: Species don’t circulate; viruses do. Please correct taxonomy and use virus names (not species names) when talking about viruses as nouns. No italics.

Line 66-67: “…some have been identified during active viral discovery programmes”. Please provide citations as examples. Also consider adding a sentence to say that some viruses have been discovered by chance, as incidental findings as part of a disease investigation. (I.e., a virus was identified in a sick person, but the virus was not thought to be the cause of the disease. Just an incidental finding).

Lines 109-111. I realize that this is likely not a fair thing to ask, but given the impact of the current pandemic, it seems that including SARS-CoV-2 would be good?

I would appreciate some justification for the taxonomic boundaries they have used, or at least some discussion about the potential limitations of this approach. While it might seem robust to use ICTV classifications – species demarcations vary widely by virus family. The factors used to demarcate species in one family can be quite different from the factors used to demarcate species in another family. I recognize that there is clearly no good way to address this problem, just as there is no clear answer for how to deal with variation in research effort. However, like the research effort problem, this has the potential to significantly impact their results. For example – is it reasonable to consider all influenza viruses as one example while separating out enteroviruses into so many species?

Figure 6. Do these hotspots indicate where good lab capacity exists? I am trying to understand the specific significance of this figure. What does it really mean and how would I use it to guide surveillance?

Reviewer #3: Abstract should include mention of investigative effort specifically (the process of virus detection) as a driver of discovery, as highlighted in the paper.

Given there is overlap between zoonotic viruses, human transmissible viruses, and vector-borne viruses, this overlap should be clarified in the text, by showing % overlap among categories, ie for human transmissible viruses, what % are zoonotic.

Information on how the geocoded location for virus detection was ascertained should be described at least briefly in the main paper, as opposed to only in the SI. When describing patient’s address, was this the location of the hospital or clinic, site of initial human exposure/infection with the virus? Were discoveries of virus in people that likely contacted the virus while travelling included? If so, how would this affect the findings?

The methods for data collection should be fully characterized, including search terms, databases searched, and inclusion or exclusion criteria, to enable repeatability. This information does not appear to be readily available in the link provided.

Line 224 - The partial dependence plots do not provide detailed descriptions of the relationships. This should be rephrased. For all figures of partial dependence plots, the x axes should be labelled. Many of these variables (eg primary land, secondary land, urbanization of primary land) are not readily understood unless the SI table is reviewed. Brief descriptions within the main text would ensure improved understanding of the main findings.

The relationship between GDP and virus discovery should be better explained, especially with respect to how the GDP of the country of virus detection and GDP change is related to detection. Is the GDP of a country where a virus was first detected generally reflect effort or resources spent toward investigative efforts at that location? Share evidence to support interpretation. Brazil, Nigeria, and Uganda are areas with high virus detection – how does GDP explain these detections and what were the specific factors involved in virus discovery; was this evenly distributed in time or based on unique efforts?

Lines 327-331 – does urbanization reflect a change in habitat? If so over what time scale? Or is this variable reflective of the level of urbanization at the time of analysis? Provide supporting evidence or documentation characterizing urbanization as a change.

Line 347 – the effect of biodiversity on virus emergence has been debated and this finding is minimally discussed here even though this is one of the effects that could be causally related to virus discovery in humans. Could the authors add more discussion of the relationship between biodiversity and virus discovery – examine this by potentially causal mechanisms and infer what this means for future trends?

The conclusion could be more forward looking, rather than a restatement of main findings already discussed

Minor editorial suggestions

Line 31: is = are

Line 89: Expand literature review methods description.

Results: start this section with a description of the major findings, not an immediate reference to a figure in the first word of the first sentence.

Line 331: rephrase, missing a verb

Line 359: awkward sentence, especially as start of new paragraph – what were other limitations mentioned?

Use of the term ‘species’ in figures and legends should be rephrased as ‘virus species’.

Figure 1: Legend should provide more detail and clarity on data points (to replace “occurrences” and define “centroids”. The 1B figure should have a descriptive y axis label.

Figure 5: revise to have a title and descriptive legend.

SI Fig. this methods overview is not especially helpful in understanding approach or methods.

S2 Fig needs a more descriptive legend. The bar plots overlap the numbers and are difficult to discern. Improve this graphic for readability and quality.

S4 Fig, S7 Fig, S9 Fig, S10 Fig; units are unclear for some plots.

S12 Fig – the units of measure are not clear for many plots. Colors on maps do not always match the range shown on the color bar. The legend does not seem to match this figure; expand description of what is being shown in the maps.

PLOS authors have the option to publish the peer review history of their article (what does this mean?). If published, this will include your full peer review and any attached files.

Reviewer #1: **Yes: **Toph Allen

Reviewer #2: No

Reviewer #3: No
---

## [Editor Report · Decision Letter 1]

19 Oct 2020

Dear Ms Zhang,

We are pleased to inform you that your manuscript 'Global discovery of human-infective RNA viruses: A modelling analysis' has been provisionally accepted for publication in PLOS Pathogens.

Best regards,

Stephen Morse

Guest Editor

PLOS Pathogens

David Wang

Section Editor

PLOS Pathogens

Kasturi Haldar

Editor-in-Chief

PLOS Pathogens

orcid.org/0000-0001-5065-158X

Michael Malim

Editor-in-Chief

PLOS Pathogens

orcid.org/0000-0002-7699-2064

Thank you for your revisions in response to the review comments.
---

## [Editor Report · Acceptance letter]

18 Nov 2020

Dear Ms Zhang,

We are delighted to inform you that your manuscript, "Global discovery of human-infective RNA viruses: A modelling analysis," has been formally accepted for publication in PLOS Pathogens.

Best regards,

Kasturi Haldar

Editor-in-Chief

PLOS Pathogens

orcid.org/0000-0001-5065-158X

Michael Malim

Editor-in-Chief

PLOS Pathogens

orcid.org/0000-0002-7699-2064